# Surveying 'Dating Violence' and Stalking Victimisation among Students at an English University: Findings and Methodological Reflections on Using a US Survey Instrument

Anna Bull [1,*] and Alexander Bradley [2]

1   Department of Education, University of York, York YO10 5DD, UK
2   School of Education and Sociology, University of Portsmouth, Portsmouth PO1 2DZ, UK; alexander.bradley@port.ac.uk
*   Correspondence: anna.bull@york.ac.uk

**Abstract:** Domestic abuse and stalking in higher education (HE) have been overlooked in research in comparison to sexual harassment and sexual violence. This article reports on survey data from 725 students at an English university using measures of stalking and 'dating violence'—physical and psychological violence from an intimate partner—from a US survey instrument (the Administrator Researcher Campus Climate Collaborative (ARC3) survey). According to this measure, 26% of respondents had been subjected to 'dating violence' and 16% to stalking behaviours. However, these findings need to be contextualised within a critical discussion about the use of the ARC3 survey tool in the English context. The ARC3 questions on 'dating violence' focus on physical and 'psychological violence'; the questions therefore omit further types of domestic abuse under UK definitions. In relation to stalking, US definitions—as captured in the ARC3 survey instrument—define specific behaviours. By contrast, in England and Wales, stalking involves behaviours that engender fear or distress in a pattern of behaviour over time. These differences mean that the ARC3 modules on stalking and 'dating violence' would need to be significantly adapted to be suitable for use in England and Wales.

**Keywords:** higher education; students; domestic abuse; stalking; gender-based violence; survey

## 1. Introduction

While in the US, many higher education institutions (HEIs) have carried out 'campus climate surveys' to understand prevalence, patterns, and attitudes towards gender-based violence and harassment (GBVH) among their student populations, such studies are not yet commonplace in many other national contexts. Across the four nations of the UK[1], amidst increasing public and policy imperatives for HEIs to do more to address GBVH (Office for Students 2021; Universities UK 2016; White 2022; Women and Equalities Select Committee 2022), there is a developing discussion of survey methods and tools in this area (Lagdon et al. 2022; Steele et al. 2021). This article aims to contribute to discussions of data collection and analysis in this area through reporting on findings and methodological challenges from a survey carried out at one English HEI in 2020. In particular, while other surveys in the UK in this area (University of Bristol SU 2021; Brook 2019; Imperial College Union 2022; Lagdon et al. 2022; National Union of Students 2010; Revolt Sexual Assault 2018; Steele et al. 2021) have prioritised sexual violence and harassment, this article focuses on the data collection instruments and findings around stalking and domestic abuse, areas which have been under-explored in higher education (HE) (DeKeseredy et al. 2017; Khan 2021). Stalking is defined here as 'a pattern of fixated and obsessive behaviour which is repeated, persistent, intrusive and causes fear of violence or engenders alarm and distress in the victim' (Suzy Lamplugh Trust 2021, p. 3), and domestic abuse is defined as 'any incident or pattern of incidents of controlling, coercive, threatening behaviour, violence

or abuse between those aged 16 or over who are, or have been, intimate partners or family members regardless of gender or sexuality' (Khan 2021; Crown Prosecution Service 2017).

This article's contribution is therefore twofold: first to report on new data in the under-researched areas of stalking and domestic abuse in HE; and second, to contribute to methodological debates around measuring domestic abuse and stalking in HE, in particular the cross-national applicability of survey tools in this area. As such, the article will be of interest to researchers and policymakers internationally who are concerned with data collection and analysis related to GBVH in HE.

The article starts via outlining the gap in research around stalking and domestic abuse in HE and introducing methodological challenges in surveys on these issues. After describing the methods, the article introduces survey findings on stalking and 'dating violence' (the term used in the survey tool). The discussion situates these findings within existing research as well as discussing limitations of the survey modules and challenges with adapting them to the context of England. Overall, the article argues that there are limitations to these measures as well as significant challenges to adapting them to context of England and Wales.

## 2. Literature Review

### 2.1. Stalking and Domestic Abuse among University Students

Both internationally and in the UK, surveys of gender-based violence victimisation have primarily focused on sexual violence and harassment. In Wood et al.'s (2017, p. 1260) overview of ten commonly used survey tools for US-based surveys on GBVH and attitudes to violence among students, all ten include sexual violence, while only seven include 'intimate partner violence' and only five include stalking. As they note, 'little is known about the types of intimate partner violence experienced by college students' (Wood et al. 2020, p. 287). In other international contexts, Heywood et al.'s (2022) national survey of university students in Australia focused on sexual violence and sexual harassment, with stalking and domestic abuse mentioned in the qualitative research but not explicitly included in the quantitative survey. Similarly, MacNeela et al. (2022, p. 245), in their national survey of university students in Ireland, include stalking within their broader definition of sexual harassment and violence but do not ask any specific questions about it.

However, there is an emerging body of work in this area. Wood et al., in a study of 6818 female students across eight US universities, found that 31% had experienced intimate partner violence (Wood et al. 2020). DeKeseredy et al. found that nearly one in five (18.2 per cent, n = 551) women in their sample reported experiencing one or more types of intimate partner violence (2019, p. 286). In a qualitative study carried out in the UK, Bull and Rye found evidence of students being subjected to stalking behaviours and domestic abuse from staff/faculty (Bull and Rye 2018); these sometimes escalated from 'grooming' and boundary-blurring behaviours (Bull and Page 2021). Studies of students' experiences of stalking, using different survey instruments and time periods (as discussed in more detail below) have found prevalence rates from 6.2% to 38% (DeKeseredy et al. 2014, p. 28; DeKeseredy et al. 2017; McCarry et al. 2021; National Union of Students 2010; Office for National Statistics 2020; Shorey et al. 2015; Speak Out Iowa 2021). Studies surveying only women students found higher rates. One study in Ghana found 80% of the 117 women students they surveyed had experienced one stalking behaviour in the last six months (Zagurny et al. 2022), while in the US, studies found 20% and 44.9% of women respondents experienced stalking while enrolled at their current institution (Buhi et al. 2009; DeKeseredy et al. 2019, p. 288).

Within UK-based research on GBVH in HE, stalking and domestic abuse are similarly under-explored (other than when they occur in the form of sexual violence or harassment, as outlined below). There exist relatively few survey-based studies examining GBVH in UK HE, but recent studies including Steele et al. (2021) and Lagdon et al. (2022) focus on sexual harassment and violence. Non-academic surveys from students' unions and activist groups have the same focus (University of Bristol SU 2021; Brook 2019; Imperial

College Union 2022; Revolt Sexual Assault 2018). There are only two published surveys, excepting the one reported here, that explicitly include findings on stalking and domestic abuse. First, the National Union of Students' Hidden Marks report included stalking as well as questions about coercive behaviours and physical violence, although not necessarily from current/former partners (National Union of Students 2010). Second, McCarry et al. (2021) surveyed staff and students across four HEIs in Scotland and included questions on stalking as well as on 'emotional abuse' and 'physical abuse' (as discussed in more detail below).

Despite this lack of explicit attention on these areas, there does exist relevant evidence in existing studies where studies have reported data about sexual violence perpetrated by romantic or intimate partners. For example, in Australia, Heywood et al. (2022, p. 40) found that 13.2% of those who had been subjected to sexual assault named a partner or ex-partner as the perpetrator of the most impactful incident they had experienced. The qualitative research accompanying this study primarily explored sexual assault and harassment, but it did also include some evidence of stalking and domestic abuse (pp. 10, 18, 22). MacNeela et al. found that 10.8% named a 'romantic partner' and 10.9% a 'former romantic partner' as the person responsible for the sexual violence situation that had the greatest effect on them (MacNeela et al. 2022, p. 219). Lagdon et al. found that 32.1% of their total sample had experienced unwanted sexual experiences from a current or previous romantic partner (2022, p. 11). Nevertheless, sexual violence is only one aspect of domestic abuse (as outlined below), and there is a dearth of data relating to economic, physical, and psychological forms of domestic abuse experienced by students.

While the study reported here uses a 'modular' approach to understanding GBVH that asks about stalking and domestic abuse separately, it is important to note that they are in fact interconnected. Outside of higher education, an analysis of stalking prosecutions in the UK during 2020 found that most offences were committed by abusive ex-partners (Crown Prosecution Service 2020). Within higher education, data from the US suggest that this pattern is not as stark among students; DeKeseredy et al. (2014, p. 28) found that eight percent of the stalking victims in their campus climate study reported that the perpetrator was a current or former intimate partner. However, other studies of stalking experienced by university students have not asked about the nature of the relationship with the person (or people) who carried out the stalking behaviours (Zagurny et al. 2022).

The lack of focus on domestic abuse and stalking in existing studies of GBVH in HE is problematic for several reasons. Firstly, young people are more at risk of both domestic abuse and stalking than older people, and therefore university students are also more likely to experience these behaviours than the general population (Khan 2021; Office for National Statistics 2020). Second, both stalking and domestic abuse put victims at risk of femicide or suicide (Khan 2021; Monckton-Smith 2021). As Khan notes, 'the impact of domestic abuse on victims is often chronic, devastating, and may be life-threatening' (Khan 2021, p. 13). Similarly, stalking can be extremely distressing, and in a recent survey, the Suzy Lamplugh Trust found the 94% of people who'd experienced stalking during lockdown said their mental health had been impacted by it (Suzy Lamplugh Trust 2021, p. 12). Overall, then, the current focus on sexual violence and harassment means that the full spectrum of GBVH is not being captured in most existing UK surveys. As Khan has noted, this reflects a wider lack of attention to domestic abuse among prevention and response work within HE (Khan 2021), despite recent steps outlining work to be done in this area (Universities UK 2020a, 2020b).

### 2.1.1. Defining Stalking and Domestic Abuse

While there is no strict legal definition of 'stalking' in Great Britain (Crown Prosecution Service 2018), in order to be prosecuted as a criminal offence, stalking needs to involve 'a course of conduct or pattern of behaviour which causes someone to fear that violence will be used against them on at least two occasions, or which causes them serious alarm or distress to the extent it has a substantial adverse effect on their day-to-day activities' (Crown

Prosecution Service 2017). Stalking may also overlap with domestic abuse (as defined below), in which people who are in abusive relationships may also be subjected to stalking behaviours from their partners. Legal definitions may not, of course, be appropriate or helpful in an HE context; as Vera-Gray and Kelly note, 'crime and victimisation surveys and legal frameworks systematically exclude forms of violence and abuse that are more likely to be experienced by women than men', an issue that stems from using 'a male as norm understanding of what counts as crime' (Vera-Gray and Kelly 2020, p. 268). Nevertheless, legal and socio-cultural understandings sometimes overlap or draw on each other. Indeed, anti-stalking charity The Suzy Lamplugh Trust define stalking 'a pattern of fixated and obsessive behaviour which is repeated, persistent, intrusive and causes fear of violence or engenders alarm and distress in the victim' (Suzy Lamplugh Trust 2021, p. 3). In the discussion below, we therefore draw on points of similarity across both definitions.

While the term 'domestic abuse' is commonly used in the UK, in the analysis and discussion below, we use the term 'dating violence' in order to reflect the wording of the ARC3 survey. These two terms should be understood as distinct. This is because, as discussed below, the ARC3 questions on 'dating violence' focus solely on psychological and physical violence and, as such, do not translate smoothly to a UK context. The Crown Prosecution Service (CPS) in England uses the following definition of domestic abuse, which is also recommended by Khan (2021) in their guidance for universities:

> Any incident or pattern of incidents of controlling, coercive, threatening behaviour, violence or abuse between those aged 16 or over who are, or have been, intimate partners or family members regardless of gender or sexuality. The abuse can encompass, but is not limited to, psychological, physical, sexual, financial and emotional. This definition includes so-called 'honour'-based violence, forced marriage, and female genital mutilation (FGM).
>
> (Khan 2021; Crown Prosecution Service 2017)

A further type of domestic abuse—'controlling, coercive' behaviour between two people who are 'personally connected'—became a criminal offence in England and Wales in 2015, in Scotland in 2018, and in Northern Ireland in 2021. A further facet of domestic abuse, 'tech-mediated abuse' is included by Universities UK, a lobby group for HE in the UK, in their definition (Universities UK 2020a, 2020b). As can be seen from the above definition, domestic abuse encompasses a range of behaviours that are challenging to capture in a short survey module.

### 2.1.2. Definitions and Methodological Challenges

There are significant methodological and conceptual challenges to surveying these behaviours. However, as the US has a relatively well-developed set of survey tools in this area (Wood et al. 2017), it makes sense to draw on this work in the UK. In particular, it is a strength of the Administrator Researcher Campus Climate Collaborative (ARC3) survey (introduced below) that it includes modules on stalking and 'dating violence'. Nevertheless, there are significant challenges to adapting surveys across contexts. For example, in legal definitions of stalking, as Purcell et al. (2004b) describe, there is no consistency across jurisdictions. For example, in different jurisdictions, stalking may or may not involve repeated events, and these events may or may not be required to bring about fear or distress. Some jurisdictions—notably in the US—codify specific behaviours that constitute stalking, while others do not. Precisely which behaviours should be included changes over time, for example, cyberstalking behaviours are now important to include (Henry et al. 2020).

In measuring domestic abuse there are even more significant methodological challenges for survey research (Myhill 2015; Myhill 2017; Walby et al. 2017). A recent review of government data collection on domestic abuse in the UK has recommended significant changes in its measurement (Office for National Statistics 2021), in particular distinguishing between different 'abuse profiles' in order to distinguish between 'situational' violence,

which is not part of a wider exertion of control, and forms of 'intimate terror' where violence occurs alongside, and in order to exert, control (Johnson 2008; Myhill 2017). Failing to distinguish between these different forms of domestic abuse can obscure the gendered patterns of domestic abuse (Myhill 2017). 'Intimate terror' has more recently been theorised as 'coercive control', which was criminalised in 2015 in the England and Wales (Crown Prosecution Service 2017). However, as the Office for National Statistics (ONS) for England and Wales notes, 'there remain significant issues relating to measuring coercive control, and there is no agreed measurement instrument internationally' (Office for National Statistics 2021). Nevertheless, the Crime Survey of England and Wales (CSEW) includes questions that have been used as an indicator for coercive control (Myhill 2015). In discussing the findings of this study, this article will therefore assess the extent to which the ARC3 survey modules are appropriate to use in the UK context, given these competing definitions as well as ongoing debates on measurement.

*2.2. Methods*

Research design is always a balance between ideal approaches and pragmatism. This study took place at a post-1992 university[2] in England via a partnership between the students' union and members of academic staff. A more detailed discussion of the governance and dissemination issues we encountered can be found in the description of 'University C' in Bull et al. (2022). The primary purpose of the survey was to provide evidence to the university of the scale of GBVH victimization among the student population in order to encourage investment from the institution to address this issue. This purpose led to research design decisions that were sub-optimal from a methodological perspective. Most notably, the partnership with the students' union meant that the research was carried out under very strict time constraints. Students' union officers in UK universities are elected for one academic year, so we had significant time limitations as a result of this partnership. We needed to get the survey through ethical approval, out to students, and reported on within the one academic year our students' union partner was in office.

In order to work within these constraints, we chose an off-the-shelf survey instrument designed for a student population that gathered victimization data across a range of types of GBVH. There were, at the time of writing, no surveys of GBVH that were devised for a UK student population. In the US, by contrast, there is an extensive tradition of campus climate surveys and multiple tools available. The ARC3 survey, which we chose for this study, is an open-access tool developed by a group of academic researchers and HE administrators in the US which has been used extensively in 'campus climate surveys' in the US (Swartout et al. 2019). It is a cross-sectional survey which comprises 19 modules covering victimisation and perpetration, as well as further modules, for example, on consent, alcohol use, and peer norms, that can be selected to fit the needs of specific institutions. For this study, four victimisation modules were used, on sexual violence, sexual harassment, 'dating violence', and stalking. The first two modules proved to be appropriate and helpful for the UK context. However, there were challenges in using the latter two modules, which are therefore discussed in this study. We also used the updated Illinois Rape Myth Acceptance scale (McMahon and Farmer 2011) and a module on professional boundaries between staff and students as well as selected demographic questions matching the study from the National Union of Students and The 1752 Group 2018 (2018).

The ARC3 was chosen for several reasons. First, it was developed by a national network in the US of academic researchers in partnership with university administrators, bringing together expertise on students and gender-based violence along with practical knowledge around carrying out surveys within HEIs. Second, it was open access and publicly available for free, with guidance notes, and we had support from one of the design team, Bill Flack. Third, it used existing validated measures, including the Sexual Experiences Survey for the sexual violence and sexual harassment modules. Fourth, these two modules had recently been used for a national survey carried out in Ireland, which

meant that this instrument would allow comparison of findings from these two modules across the UK, Ireland, and US institutions that had used this tool. Fifth, the US has sufficient cultural and linguistic similarity to the UK that minimal adaptation was required. Finally, its modular structure meant that we could choose which modules to include and adapt it to our local needs in this way.

The ARC3 was chosen over UK-specific surveys because there did not appear to be any youth-specific tools that encompassed a wider spectrum of forms of GBVH including stalking and domestic abuse. As outlined below, development work is ongoing within the Office for National Statistics to devise a survey instrument to better measure domestic abuse in a way that includes coercive control, but this was not available at the time the study was being carried out. Other international survey tools, including those devised in the Global South such as the WHO Violence Against Women Instrument, have the advantage of covering a range of forms of abuse but are aimed at women who are married/partnered rather than students who may be participating in 'hook-up' cultures. Due to the distinctive cultures of undergraduate-age university students (who made up the majority of the student population at the institution in question), we felt it was important to use a survey tool that had been designed for, and piloted in, higher education in order to pay attention to student sexual cultures, which have some common factors across the UK and US (see for example Jackson and Sundaram 2020; Hirsch and Khan 2020).

We worked with our students' union partners to review the wording of the questions to ensure they were appropriate for the specific population, changing a few terms (for example, removing the word 'horseplay' but leaving 'joking around'). We then ran the survey with minor adaptations. Due to time constraints, we did not carry out a pilot on this specific population.

### 2.2.1. Participants

The survey was sent to all students enrolled at the university (N = 31,059) in November 2020 via email from the Students' Union Welfare Officer. 1303 students filled out the survey (response rate = 4.19%) over a three-week period, and 725 of these consented to their responses being analysed and reported on publicly.[3] All participants who completed the survey were offered the opportunity of entering into a raffle to win a £50 Amazon voucher. Of the 725 respondents whose data are reported here, 62% were women, 33% were men, 3% non-binary, and 2% preferred not to disclose (see Table 1). The majority of respondents were between 18 and 24 (83%) years old, were UK-domiciled or 'home' students (84%), and were studying at undergraduate level (85%).

**Table 1.** Demographic characteristics of the sample.

| Demographics of Sample | n | % of Sample |
|---|---|---|
| Gender | | |
| Women | 446 | 62 |
| Men | 238 | 33 |
| Non-Binary | 20 | 3 |
| Prefer not to say | 14 | 2 |
| Other | 5 | 0 |
| Age | | |
| 18–24 | 601 | 83 |
| 25–29 | 58 | 8 |
| 30–39 | 41 | 6 |
| 40–49 | 15 | 2 |

**Table 1.** *Cont*.

| Demographics of Sample | n | % of Sample |
|---|---|---|
| 50–59 | 5 | 1 |
| 60–65 | 1 | 0 |
| Over 65 | 2 | 0 |
| UK-domiciled ('home') or international students | | |
| UK-domiciled ('home') | 602 | 84 |
| International | 117 | 16 |
| Level of Study | | |
| First Years | 300 | 42 |
| Second Year | 159 | 22 |
| Third Year | 121 | 17 |
| Fourth Year | 23 | 3 |
| Placement Student | 7 | 1 |
| Masters Year | 84 | 12 |
| PhD | 10 | 1 |
| Other | 17 | 2 |

### 2.2.2. Procedure

After initially being due to be rolled out in March 2020, the survey was delayed due to the outbreak of the COVID-19 pandemic. In line with advice from US experts (Holland et al. 2020), rather than delaying until after the pandemic, the survey was eventually rolled out in November–December 2020, while a second national lockdown in England was in place (from 5th November onwards). As a result, the first-year undergraduate students who responded to the survey had only experienced around six weeks of university out of lockdown by the time they filled out this survey.

### 2.2.3. Instruments

Stalking victimization (discussed in more detail below) was captured with 10 items on a 5-point scale from none (0) to more than 8 (4) ($\alpha = 0.83$). Response categories were None; 1–2; 3–5; 6–8; or more than 8 times. The 'dating violence' scale (also discussed below) contained 6 items measured on a 5-point scale from never (0) to many times (4) ($\alpha = 0.74$). Responses included 'Never; Once or twice; Sometimes; Often; Many times' (see ARC3 'technical guidance' for more details). Each measure was followed by four items designed to capture perpetrator characteristics. The four items were the gender of the perpetrator, whether the perpetrator was a student, the relationship between the perpetrator and the victim, and whether the incident/s took place on campus.

### 2.2.4. Ethics

The project gained a favourable ethical opinion from the university's Faculty of Social Sciences and Humanities ethics committee before the survey was distributed. While permission was given from the ethics review panel to name the institution, subsequent difficulties in reporting the findings (as outlined in Bull et al. 2022) mean that here we have anonymised the institution as 'University of X'. Respondents were signposted to support services both inside and outside the university at the start and the end of the survey, and the survey team liaised with the university's Wellbeing Service to arrange proactive support.

### 2.2.5. Data Analysis

Analysis of the data reported on below aimed to answer the following questions:

1. What proportion of respondents had experienced 'dating violence' and stalking since enrolling at the University of X?
2. Which students were most likely to experience these harms?
3. What factors were associated with experiencing stalking and 'dating violence', in terms of the likelihood of experiencing these harms and frequency of experience?

In order to assess the proportion of respondents who had experienced stalking and 'dating violence', we used the percentages of individual items, means, and medians from the sum score of the modules designed to capture stalking and 'dating violence'. In this study, we defined a stalking victim as anyone who had experienced one stalking event 6–8 times, three stalking events happening 1–2 times, or a combination of one stalking event 1–2 times and one 3–5 times. This means that for someone to be defined as having experienced stalking, they had to have experienced at least three stalking behaviours. We then used sum scores of these modules along with demographic factors (gender, level of study, and 'home' versus international students) to explore which students were most likely to experience these harms.

We used a logistic regression to explore how certain characteristics (gender, level of study, home vs. international status) impacted the likelihood of experiencing stalking and 'dating violence'. We then used a negative binomial regression to explore whether those characteristics also predicted the frequency with which stalking victimisation and dating violence occurred. Negative binomial regressions were chosen given the right-skewed nature of the frequency distributions for 'dating violence' and stalking victimisation.

## 3. Findings

The wider findings of the survey show that sexual or gender harassment had been experienced by 55% of respondents since enrolling at the University of X (Bull and Turner-McIntyre 2023a, 2023b). Thirty percent of respondents had been subjected to sexual violence since enrolling. Women and non-binary students were more likely than men to experience harassment or sexual violence, and they also experienced this more often than men. Eighty-three percent of sexual and gender harassment was carried out by another student studying at the university. Similarly, other students at the university were named as the person who carried out 82% of reported sexual violence incidents, 70% of all stalking victimization, and 65% of dating violence incidents reported. The survey also explored two attitudinal areas: rape myth acceptance and students' attitudes towards professional boundaries with staff (Bull et al. 2023). Below, we focus on the findings on stalking and on 'dating violence' in turn.

### 3.1. Findings on Stalking Victimization

First of all, we looked at whether respondents had ever experienced stalking behaviours since enrolling as a student at the University of X. Using the threshold described above of experiencing at least three stalking events since enrolling at the university, 16% (119 out of 725) had experienced stalking. The three most common forms of stalking victimisation experienced by students were receiving unwanted emails, instant messages, or social media messages (17%); receiving rude/mean online comments (15%); and having been left unwanted notes, texts, or voice messages (15%) (see Figure 1), all of which occurred at least one or more times. Twelve percent reported at least one experience or more of being watched, followed, or spied upon from a distance or with listening device/camera or GPS. Twelve percent also reported at least one experience of being approached at home, work, or school when they did not wish the person to be there.

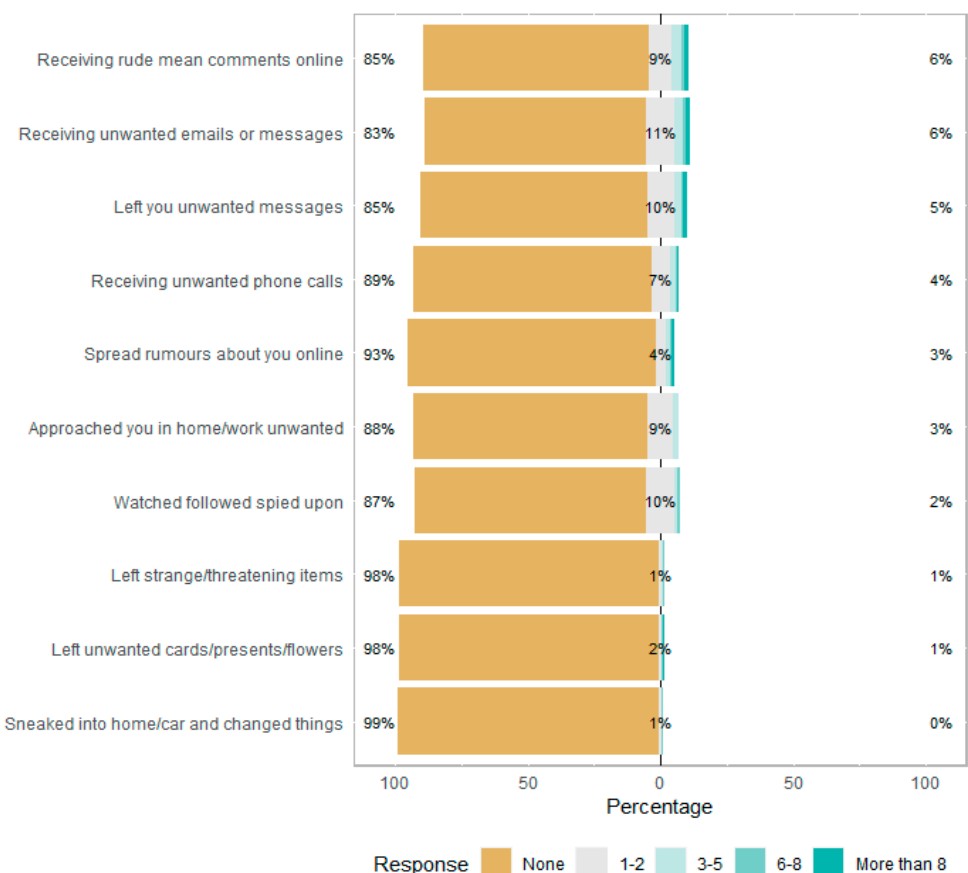

**Figure 1.** Frequency of the different types of stalking experienced by students.

Both logistic and negative binomial regression were performed to identify factors that increased students' likelihood and frequency of having experienced stalking behaviours (see Table 2). Women were more likely than men to have a higher likelihood and frequency of experiencing the behaviours described (likelihood: odds ratio (OR) = 1.85, SE = 0.25, $p < 0.05$; frequency: incidence rate ratio (IRR) = 1.53, $p < 0.05$). Non-binary students were more likely to have experienced stalking compared to men (OR = 4.03, SE = 0.55, $p < 0.05$); however, there was no difference in the frequency (IRR = 1.88, SE = 0.51, $p = 0.21$) of experiencing stalking. International students were less likely to have experienced stalking compared to home students (OR = 0.50, SE = 0.35, $p < 0.05$), but there was no difference in the frequency of stalking experienced (IRR = 0.76, SE = 0.24, $p = 0.26$). Second-year (OR = 2.52, SE = 0.27, $p < 0.001$), third-year (OR = 2.45, SE = 0.29, $p < 0.01$), and PhD (OR = 4.31, SE = 0.73, $p < 0.05$) students were more likely to have experienced stalking victimisation compared to first years, as well as to have experienced higher incidences of stalking victimisation (Second IRR = 2.29, SE = 0.22, $p < 0.001$; Third IRR = 1.80, SE = 0.24, $p < 0.05$; PhD IRR = 4.24, SE = 0.48, $p < 0.05$). As the survey asked about all experiences since enrolling at this university, some of this increase is likely to be due to second-year students having been at university for longer.

**Table 2.** Logistic regression and negative binomial regression showing factors impacting likelihood and frequency of experiencing stalking victimisation.

| Logistic Regression | Odds Ratio (95% CI) | SE | *p* |
|---|---|---|---|
| Gender: Women | 1.85 (1.13–3.02) | 0.25 | <0.05 * |
| Gender: Non-Binary | 4.03 (1.36–11.87) | 0.55 | <0.05 * |
| Gender: Not disclosed | 0.00 (0.00–∞) | 626.27 | =0.98 |
| Gender: Other | 5.22 (0.71–38.30) | 1.02 | =0.10 |
| Home/International | 0.50 (0.25–0.99) | 0.35 | <0.05 * |
| Year 2 | 2.52 (1.49–4.26) | 0.27 | <0.001 *** |
| Year 3 | 2.45 (1.40–4.30) | 0.29 | <0.01 ** |
| Year 4 | 2.45 (0.84–7.15) | 0.55 | =0.10 |
| Placement | 4.86 (0.96–24.59) | 0.83 | =0.06 |
| Masters | 1.06 (0.46–2.45) | 0.42 | =0.88 |
| PhD | 4.31 (1.03–18.01) | 0.73 | <0.05 * |
| Other Level Study | 5.50 (0.07–4.32) | 1.05 | =0.57 |
| **Negative Binomial Regression** | **Incident Rate Ratios (95% CI)** | **SE** | **p** |
| Gender: Women | 1.53 (1.06–2.20) | 0.19 | <0.05 * |
| Gender: Non-Binary | 1.88 (0.69–5.12) | 0.51 | =0.21 |
| Gender: Not disclosed | 0.07 (0.01–0.80) | 1.23 | <0.05 * |
| Gender: Other | 1.52 (0.21–10.91) | 1.00 | =0.67 |
| Home/International | 0.76 (0.48–1.22) | 0.24 | =0.26 |
| Year 2 | 2.29 (1.50–3.52) | 0.22 | <0.001 *** |
| Year 3 | 1.80 (1.13–2.88) | 0.24 | <0.05 * |
| Year 4 | 2.44 (0.95–6.31) | 0.48 | =0.06 |
| Placement | 3.80 (0.78–18.58) | 0.81 | =0.10 |
| Masters | 1.02 (0.57–1.84) | 0.30 | =0.94 |
| PhD | 4.24 (1.13–15.91) | 0.68 | <0.05 * |
| Other Level Study | 0.69 (0.21–2.21) | 0.60 | =0.53 |

Note 1. Logistic regression likelihood test $\chi^2$ (12) = 45.99, $p < 0.001$. * refers to probability level $p < 0.05$, ** relates to probability level $p < 0.01$ and *** refers to probability less than $p < 0.001$.

### 3.2. Findings on 'Dating Violence'

'Dating violence' was experienced by a large minority of respondents (see Figure 2). Physical experiences of dating violence included 11% (n = 78) of respondents reporting experiencing at least one experience of being pushed, grabbed, or shook by someone they had been in a relationship with during their time at the university; 4% (n = 29) reported one or more experiences of being hit; and 1% (n = 7) reported being physically beaten up one or more times. In relation to psychological experiences of dating violence, in response to the question 'Not including joking around, the person threatened to hurt me and I thought I might get really hurt', 16% (n = 118) of respondents reported at least one experience. Eight percent (n = 57) reported being threatened. As Figure 2 shows, 'psychological' experiences of dating violence were more common than physical violence. Some students were subjected to these behaviours more than once. This included 2% (n = 15) of those who received threats of harm, and 1% (n = 7) of those who were hit.

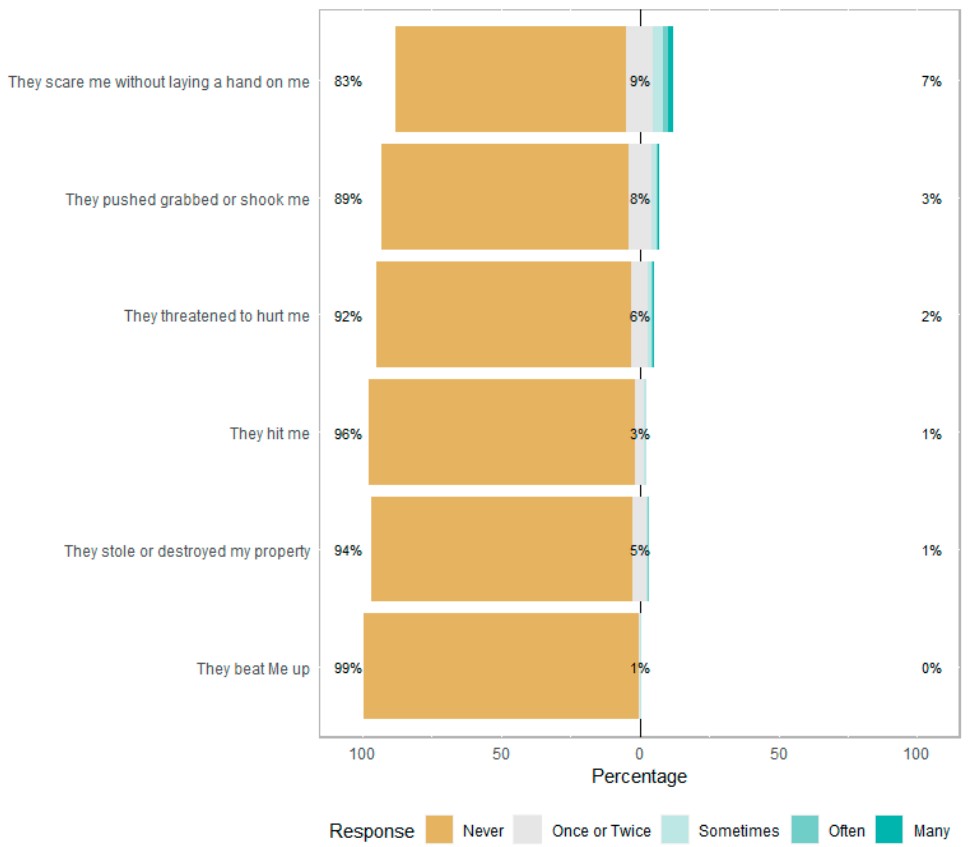

**Figure 2.** Frequency of the different types of 'dating violence' experienced by students.

Women students were much more at risk of dating violence than men students. Overall, a quarter (26%) of students who completed the survey had experienced at least one behaviour associated with dating violence, with 15% experiencing some form of physical dating violence and 23% experiencing some form of psychological dating violence. Women were 1.49 times more likely to have experienced dating violence compared to men (See Table 3; OR = 1.49, SE = 0.20, *p* < 0.05) and experienced a higher frequency of incidents compared to men (IRR = 2.01, SE = 0.24, *p* < 0.001). Non-binary people were four times as likely to have experienced dating violence (OR = 4.08, SE = 0.48, *p* < 0.01) and reported higher incidences of dating violence than men (IRR = 2.54, SE = 0.24, *p* < 0.01). Second-year students were more likely to have experienced dating violence compared to first years (OR = 1.67, SE = 0.22, *p* < 0.05), although we do not see an effect on higher-frequency incidences (IRR = 1.23, SE = 0.27, *p* = 0.44). As above, this is likely to be due to second-year students having been at university for longer. No other effects of year of study or international/home status were found.

**Table 3.** Logistic regression and negative binomial regression showing factors impacting likelihood and frequency of experiencing dating violence.

| Logistic Regression | Odds Ratio (95% CI) | SE | *p* |
|---|---|---|---|
| Gender: Women | 1.49 (1.01–2.18) | 0.20 | <0.05 * |
| Gender: Non-Binary | 4.08 (1.59–10.51) | 0.48 | <0.01 ** |
| Gender: Not disclosed | 0.65 (0.14–3.03) | 0.79 | =0.58 |
| Gender: Other | 2.69 (0.42–17.26) | 0.95 | =0.30 |
| Home/International | 0.80 (0.49–1.31) | 0.25 | =0.37 |

**Table 3.** *Cont.*

| Logistic Regression | Odds Ratio (95% CI) | SE | *p* |
|---|---|---|---|
| Year 2 | 1.67 (1.08–2.58) | 0.22 | <0.05 * |
| Year 3 | 1.44 (0.89–0.233) | 0.25 | =0.14 |
| Year 4 | 1.99 (0.80–4.94) | 0.46 | =0.14 |
| Placement | 1.21 (0.22–6.72) | 0.87 | =0.82 |
| Masters | 1.17 (0.65–2.14) | 0.30 | =0.60 |
| PhD | 1.63 (0.41–6.55) | 0.71 | =0.49 |
| Other Level Study | 0.24 (0.03–1.81) | 1.04 | =0.16 |
| **Negative Binomial Regression** | **Incident Rate Ratios (95% CI)** | **SE** | ***p*** |
| Gender: Women | 2.01 (1.27–3.20) | 0.24 | <0.001 *** |
| Gender: Non-Binary | 2.54 (0.74–8.80) | 0.24 | <0.01 ** |
| Gender: Not disclosed | 0.89 (0.18–4.33) | 0.63 | =0.13 |
| Gender: Other | 2.92 (0.27–31.26) | 1.21 | =0.37 |
| Home/International | 0.80 (0.44–1.43) | 0.30 | =0.44 |
| Year 2 | 1.23 (0.72–2.09) | 0.27 | =0.44 |
| Year 3 | 0.93 (0.51–1.67) | 0.30 | =0.80 |
| Year 4 | 1.72 (0.56–5.24) | 0.56 | =0.34 |
| Placement | 0.49 (0.05–4.48) | 1.13 | =0.53 |
| Masters | 0.66 (0.32–1.39) | 0.38 | =0.28 |
| PhD | 2.58 (0.51–12.97) | 0.82 | =0.25 |
| Other Level Study | 0.33 (0.07–1.65) | 0.82 | =0.18 |

Note 2: Logistic regression likelihood test $\chi^2$ (12) = 24.40, $p < 0.05$. * refers to probability level $p < 0.05$, ** relates to probability level $p < 0.01$ and *** refers to probability less than $p < 0.001$.

## 4. Discussion

### 4.1. Stalking

4.1.1. Limitations of Stalking Survey Module

Before discussing these findings in relation to existing literature, there are some significant limitations in relation to the survey questions that need to be considered. The ARC3 module on stalking asks about specific behaviours that constitute stalking, following the US legal framework, which aims to avoid 'vagueness' and thus tends to specify behaviours (Purcell et al. 2004b, p. 161) rather than drawing on the broader definitions used in the UK. The wording of the ARC3 module therefore reflects this context through focusing on behaviours rather than the response of the person targeted, as follows:

Since you enrolled at the University of X, have you been in a situation when someone

1.  Watched or followed you from a distance, or spied on you with a listening device, camera, or GPS (global positioning system)?
2.  Approached you or showed up in places, such as your home, workplace, or school when you didn't want them to be there?
3.  Left strange or potentially threatening items for you to find?
4.  Sneaked into your home or car and did things to scare you by letting you know they had been there?
5.  Left you unwanted messages, such as notes, text or voice messages?
6.  Made unwanted phone calls to you (including hang up calls)?
7.  Sent you unwanted emails, instant messages, or sent messages through social media apps?
8.  Left you cards, letters, flowers, or presents when they knew you didn't want them to?



9.    Made rude or mean comments to you online?

10.    Spread rumours about you online, whether they were true or not?

These questions do not fit the more victim-centred definitions of stalking used in the UK. Although questions three and four refer to 'threatening' items and doing things 'to scare you', the module focuses on the intentions and actions of the perpetrator rather than the response of the victim. By contrast, as noted above, in the UK, stalking is defined as a pattern of behaviour causing distress, fear, or alarm (Crown Prosecution Service 2018; Suzy Lamplugh Trust 2021, p. 3). Furthermore, the language here could be updated to reflect the UK context and changes in language around technology, such as using the term 'tracker' rather than 'GPS'.

There were also other difficulties with analysing this module. First, with this survey tool, it was not possible to ascertain whether multiple events were carried out by the same person or group of people (therefore constituting a course of conduct). Second, it was not possible to ascertain whether several events occurred within a specific time period. This is important because Purcell et al. found that stalking events that continue for more than a period of two weeks are 'associated with a more intrusive, threatening and psychologically damaging course of harassment' (2004a, p. 571), so the period of time as well as the number of events is relevant. Finally, these questions could be updated to reflect the increasingly digitally mediated lives of students.

As a result of these issues, there were some difficult decisions to be made around how to analyse these data. A dichotomous approach of two categories, where one group has never experienced any of these behaviours, and a further group has experienced at least one, might lead to over-reporting as well as a mismatch with the UK context, which requires a 'course of conduct'. As noted above, we dealt with these issues through defining stalking victimisation as having experienced one stalking event 6–8 times, three stalking events happening 1–2 times, or a combination of one stalking event 1–2 times and one 3–5 times. However, it is possible that someone experienced three different events from different people over three different years, and this would be defined as stalking within this survey even though it would not constitute a 'pattern of behaviour' as required by UK definitions. Furthermore, this tool did not collect data about at the period of time within which these events happened, which is important for determining whether a pattern of intrusions can be considered as stalking (Purcell et al. 2004a).

4.1.2. Discussion of Stalking Findings

This study found that women were more likely than men to experience stalking behaviours, and for these to occur more frequently. This finding is in contrast with Shorey et al.'s US study of stalking within students' dating relationships, which found that 'stalking perpetration and victimization in current dating relationships appears to be a gender-neutral problem' (Shorey et al. 2015, p. 939). However, it is in line with McCarry et al.'s findings that stalking victimisation among students was gendered (McCarry et al. 2021, p. 34). It is possible that women experience more stalking behaviours outside of dating relationships, from former partners, acquaintances, friends, or strangers, rather than from current intimate partners. Not only this, but as Lazarus et al. found, women perceive 'psychosocial cybercrime' such as online stalking behaviours as more 'severe' than men do (Lazarus et al. 2022, p. 392). This underlines the gendered patterns of victimisation occurring here.

In line with existing discussions of cyberstalking and online harassment among young people (Henry et al. 2020; Henry and Powell 2016), digitally mediated behaviours such as receiving unwanted messages were the most likely to be reported by respondents. The prevalence, at 16%, sits within the range of prevalence obtained in other studies of students' experiences of stalking. The Hidden Marks (National Union of Students 2010) study, which also asked about experiences 'since enrolling at this institution', found that 12% of students self-reported as victims of stalking. However, Hidden Marks also included the term 'repeatedly' in the question, and referred to behaviours 'that seemed

obsessive or made you afraid or concerned for your safety' (National Union of Students 2010, p. 15). Therefore, a lower prevalence would be expected as Hidden Marks only included behaviours that engendered fear in the target. By comparison, the ONS (Office for National Statistics 2020) and McCarry et al. (2021) asked about victimisation experiences only within the last 12 months. These two studies reached very different figures: ONS found that 6.2% of respondents in England and Wales aged 16 and over who reported themselves as students had experienced stalking, while McCarry et al. found 22.8% of their student sample (in Scotland) had experienced stalking. Internationally, the range is even greater. DeKeseredy et al. (2014, p. 28) found 38% of their sample of over 5000 students had experienced at least one stalking behaviour, while a study of 4268 students at the University of Iowa (Speak Out Iowa 2021) found that 18.8% of students reported experiencing at least one stalking behaviour that seemed obsessive or made them fearful, and 8.2% of students experienced repeated stalking behaviour.

These differences demonstrate the wide range of methods that are in use for conceptualising and measuring stalking in HE. They also show that asking about repeated behaviours, or behaviours that lead to fear, make a clear difference to findings. In light of this discussion, we would suggest that this module needs to be significantly adapted to be appropriate for use in UK HEIs in order to fit within the UK context of a 'pattern of behaviour' that leads to fear. First, a set of follow-up questions could ascertain whether the events reported caused 'alarm or distress' to the respondent. Surveys in both the UK and in the US already ask about whether events led to fear or distress (McCarry et al. 2021; Tjaden and Thoennes 1998, p. 17). Second, other follow-up questions that could be tested are whether stalking events took place within a specific time period and whether they were carried out by the same person/group of people (see methods in Purcell et al. 2004a, p. 574). Alternatively, the ONS draws on the England and Wales definition of events involving 'fear, alarm or distress' as well as repeated behaviours in order to assesses stalking victimisation within six questions (Kantar Public & Office for National Statistics 2015). These questions could also be appropriate to HE. Third, questions could be updated to reflect students' increasingly digitally based lives. Fourth, we suggest a discussion of shared analysis frameworks around how many intrusions constitute victimisation (Purcell et al. 2004a, p. 574).

### 4.2. 'Dating Violence'

### 4.2.1. Limitations of 'Dating Violence' Survey Module

In the ARC3 module on 'dating violence', questions focus on psychological and physical dating violence, as follows:

This section asks about your experiences about behaviour in your relationships since you enrolled at the University of X. This includes any hook-up, boyfriend, girlfriend, husband or wife you have had, including exes, regardless the length of the relationship.

1. Not including joking around, the person threatened to hurt me and I thought I might get really hurt
2. Not including joking around, the person pushed, grabbed, or shook me
3. Not including joking around, the person hit me
4. Not including joking around, the person beat me up
5. Not including joking around, the person stole or destroyed my property
6. Not including joking around, the person can scare me without laying a hand on me

A strength of these questions is that two of them capture fear. This fits in with the UK Crown Prosecution Service's (2017) guidance on coercive control, which states that '[the behaviour] must have a 'serious effect' on someone and one way of proving this is that it causes someone to fear, on at least two occasions, that violence will be used against them'. However, they could go further in capturing a wider spectrum of coercive and controlling behaviour, including sexual and economic abuse, as covered in the UK CPS definition. Furthermore, half the questions focus on physical violence, which risks perpetuating the notion that abusive relationships are primarily about physical violence, when abusive relationships may rarely or never include physical violence (Johnson 2008). In addition,

as outlined above (Myhill 2015; Myhill 2017; Johnson 2008), there is a risk that, through including such a strong focus on physical violence, the gendered patterns of domestic abuse are not captured.

A limitation of this module is that it does not capture coercive control, having been devised before coercive control was recognised as a significant aspect of domestic abuse. Across all four UK nations, coercive control is now criminalised, and work is underway to devise appropriate survey questions to capture it (Office for National Statistics 2021). Nevertheless, questions one and six in the module could potentially capture 'generalized fear' (Myhill 2015, p. 370), which has been identified as a consequence of coercive control. By contrast, a potential weakness of the ARC3 'dating violence' module is the focus on physical violence. Myhill (2017) suggests that studies that foreground physical violence are likely to obscure gendered patterns of domestic abuse. While this study did find a gendered pattern, with women and non-binary people being more likely to report having been victimised than men, this may still be under-representing the gendered patterns.

In terms of specific behaviours that are included, it would make sense to include non-fatal strangulation, which as Edwards and Douglas (Edwards and Douglas 2021) have outlined, is both a prevalent, gendered, and highly risky behaviour associated with domestic abuse and is common among students (Herbenick et al. 2021). It has also recently become a crime in England and Wales.

To summarise, in the current iteration of the ARC3 survey, gendered patterns might be lost as respondents might report physical violence that is 'situational' rather than part of an ongoing context of fear and control (Myhill 2015; Myhill 2017). Some of these issues could be addressed in the analysis, for example, thorugh analysing the physical violence questions alongside the psychological violence questions. However, for future surveys, questions that 'focus on perpetrators' controlling tactics and behaviors' (Myhill 2015, p. 360) are needed.

### 4.2.2. Discussion of 'Dating Violence' Findings

This study found a higher prevalence of 'dating violence' than national data on domestic abuse from the Crime Survey for England and Wales, wherein full-time students have been found to be the most likely to experience domestic abuse compared to any other occupation at 7.7% of the population, with women students (11%) more than twice as likely as men students (5%) to be victimised (Khan 2021, p. 24; Office for National Statistics 2018). These differences may be due to these questions measuring 'dating violence' differently from existing survey tools for domestic abuse in the UK, most notably the CSEW, and due to the CSEW omitting students in university accommodation from its sampling (Tilley and Tseloni 2016, p. 83), but including 16- and 17-year-olds. Nevertheless, the ARC3 study includes a narrower range of questions than the CSEW, asking solely about physical and psychological violence, rather than other forms of domestic abuse such as financial or sexual abuse (Kantar Public & Office for National Statistics 2015). There is also a discrepancy in the study, whereby 18% of respondents named 'strangers' as the perpetrators of 'dating violence'. This could indicate that respondents have not followed the instruction to discuss 'experiences about behaviour in your relationships'. Alternatively, respondents could be referring to 'hook-ups', as mentioned in the question, where students have had a one-off sexual encounter with a stranger. Either way, these differences in victimisation rates across different studies require further exploration.

### 4.3. Wider Limitations of the Study

As well as the limitations specific to the survey modules discussed above, this study has several further limitations. First, data were collected during a COVID-19 lockdown in late 2020. As a result, it is possible that the findings would have been different if they had been carried out in a non-COVID year. Second, as outlined above, due to time constraints in designing and delivering the study resulting from the partnership with the students' union that facilitated this survey, the survey tool was used 'off the shelf' rather than being

adapted to the UK context before use, other than some small changes to the wording. Nor was it piloted with the specific study population. Third, further data collection, particularly on a wider range of demographic characteristics of students, would have allowed a more sophisticated analysis. Finally, the response rate was low (4.19%), which has implications for the generalizability of the findings. However, Jeffrey et al. 'found no evidence that a low response rate campus climate survey biases sexual violence victimization or perpetration rates' (Jeffrey et al. 2022, p. 549). It is reasonable to assume that the same could be said for domestic abuse and stalking, and indeed, our findings are within expected prevalence rates.

## 5. Conclusions

This article has reported on findings from a survey of 725 students at a university in England using the ARC3 survey modules on stalking and 'dating violence' victimisation. These areas have both been neglected in comparison to studies of sexual violence and harassment among student populations. In response to the module on 'dating violence' victimisation, which includes questions on psychological and physical 'dating violence', 26% of respondents had experienced one or more of the behaviours surveyed since enrolling at this institution. 'Psychological' experiences of dating violence were more common than physical violence. The prevalence, at 26% of this sample, is much higher than the 7.7% among students that was found by the Crime Survey of England and Wales (Office for National Statistics 2020). This could be due to differences in sampling, even though the CSEW uses a broader range of questions encompassing financial, sexual, physical, and psychological abuse as well as controlling behaviours (Office for National Statistics 2018, pp. 255–8). The module on stalking, using a threshold of experiencing at least three stalking events since enrolling at the university, found that 16% (119 out of 725) had experienced this. Other studies have found between 12% and 38% of students have been subjected to stalking behaviours, so this finding sits within the existing range of prevalence. Cyberstalking behaviours such as receiving unwanted messages online were the most likely type of stalking behaviour to be reported by respondents. For both stalking and 'dating violence', women and non-binary students were more likely to be victimised than men students. In relation to year of study, second-year undergraduate students were most likely to report having experienced stalking or 'dating violence' behaviours since enrolling at university.

The article has also critically assessed the appropriateness of the ARC3 modules on stalking and 'dating violence' victimisation for use in the UK context, specifically focusing on England and Wales. While the modules on sexual harassment and sexual violence victimisation from ARC3 used for this study worked well, there were significant difficulties in adapting the stalking and 'dating violence' modules to the UK. In relation to the module on stalking, the ARC3 questions focus on stalking behaviours, as fits the US legal definition of stalking. However, in the UK, both social and legal definitions require stalking to constitute a pattern of behaviour that engenders fear or distress in the victim. This means that survey instruments need to ask whether behaviours engender fear or distress, as well as capturing a pattern of behaviour over time. More generally, it is argued that a greater focus on cyberstalking is now needed in order to reflect the increasing digital mediation of students' lives.

In relation to the 'dating violence' module, the ARC3 questions focus on physical and 'psychological violence'. This focus does not capture the full spectrum of behaviours that constitute domestic abuse according to UK definitions. Furthermore, across the four nations of the UK, 'coercive control' has been criminalised, and the concept is also widely used by practitioners and researchers. Therefore, it may be more appropriate to use survey instruments that cover the full spectrum of controlling behaviours (see for example Graham-Kevan and Archer 2003). However, some of the experiences that constitute domestic abuse are covered in the ARC3 modules on stalking, sexual harassment, and sexual violence. Therefore, in order to fully capture domestic abuse within the ARC3 survey, data analysis would need to be carried out across modules in order to include all forms of domestic abuse by a current or former partner.

This issue raises the question of whether a 'modular' approach is appropriate for studying gender-based violence and harassment. On a practical level, such an approach means that relevant responses might be included in two different modules, for example, a respondent could report the same incident under both the 'harassment' module and the 'stalking' module, which may lead to double counting or inclusion in the wrong category. On a conceptual level, as Kelly has outlined, sexual violence constitutes a 'continuum' where it is not necessarily possible to categorise events discretely, and there is a 'common character' to different types of events (Kelly 1988, p. 76). Such a conceptualisation is also possible for GBVH more generally. One way forward for such studies could be to include scales that explore the continuum of GBVH and then ask about the context in which these behaviours occur in order to define experiences as 'stalking', 'domestic abuse', etc. at the analysis stage (rather than at the survey design and data collection stage, as ARC3 does). A promising example of such an approach comes from DeKeseredy et al. (2019).

This article has pointed out limitations for using the modules on stalking and 'dating violence' from the ARC3 survey instrument in the HE context of England and Wales. This critique is intended to build on the ground-breaking work of the ARC3 team in order to push forward debates in the UK and internationally on surveying GBVH in HE. It is important to emphasise that researchers and HEI leaders should not avoid gathering data on GBVH due to the absence of appropriate survey tools. In the short term, survey questions from the CSEW could be used to assess stalking behaviours in HE, adding in questions about accommodation type in order aid comparison with existing CSEW data (which omits residents in student accommodation). Once ongoing work has been completed on CSEW questions on coercive control, these may also be able to be adapted to a short-form version for use in HE. For individual institutions that are attempting to better understand their own populations to inform prevention and response efforts, a wider range of behaviours could be captured, as well as 'domestic abuse myth acceptance' (Fenton and Jones 2017). However, as this article has outlined, there is still a significant amount of work to be done to generate appropriate data collection instruments in this area.

**Author Contributions:** Conceptualization, A.B. (Anna Bull); methodology, A.B. (Alexander Bradley) and A.B. (Anna Bull); formal analysis, A.B. (Alexander Bradley); data curation, A.B. (Alexander Bradley) and A.B. (Anna Bull); writing—original draft preparation, A.B. (Anna Bull); writing—review and editing, A.B. (Alexander Bradley) and A.B. (Anna Bull); visualization, A.B. (Alexander Bradley). All authors have read and agreed to the published version of the manuscript.

**Funding:** This research received no external funding.

**Institutional Review Board Statement:** The name of the university where ethical approval was obtained is not included to enable it to be anonymised, but further details on ethical review can be obtained from the first author.

**Informed Consent Statement:** Informed consent was obtained from all subjects involved in the study.

**Data Availability Statement:** Data are available on request due to restrictions, e.g., privacy or ethical. The data presented in this study are available on request from the corresponding author. The data are not publicly available due to insufficient permissions for sharing.

**Acknowledgments:** Thanks to Hayley Turner-McIntyre for the support that made this study happen. Thanks to Emma Short from De Montfort University for advice on literature and methods for measuring stalking, and to four anonymous reviewers for extremely helpful comments.

**Conflicts of Interest:** The authors declare no conflict of interest.

## Notes

[1] The four nations of the UK (England, Wales, Scotland, and Northern Ireland) each have different regulatory regimes for higher education. Furthermore, there are three separate criminal justice systems (for England and Wales, Scotland, and Northern Ireland). This study primarily discusses England. However, in some cases, the discussion is also relevant to Great Britain (England, Wales, and Scotland) or to the whole of the UK. Therefore, at different points in the article, we have purposefully discussed 'the UK', 'Great Britain, 'England', or 'England and Wales'.

2   'Post-1992' is a designation used to describe newer universities in the UK that were given university status through legislation passed in 1992. Many of these universities had previously been other types of HEIs such as teacher training colleges or technical colleges.

3   In this study, therefore, we report on data from the 725 respondents who gave consent for their data to be publicly reported on. Comparisons of the two datasets showed no significant differences between them. The full dataset was used for an initial report to the university.

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
