# Peer review of "Surveying ‘Dating Violence’ and Stalking Victimisation among Students at an English University: Findings and Methodological Reflections on Using a US Survey Instrument"

_socsci, doi:10.3390/socsci12100561_

Round 1
Reviewer 1 Report
There is a lack of clarity to compare the concept of domestic violence and gender based violence as many countries in the EU consider that the first one is a way to hide from violence against women. You cannot compare USA and UK cultural bases as they have differences in the legislation about violence and you cannot transfer one tool to another country. You write gender when your should say sex. See Kate Millett, Sexual Politics. You say risk about women or girls without explaining that we are living in a patriarchal society near to neofascist government as is the USA. (See also How Fascism Works, Jason Stanley) The politics of Us and THEM. Men are mostly offenders and women are victims. what do you really mean with "risk"?
Line 107 gov-
ernment, is this rigth?
Reviewer 2 Report
There is a great need globally for research on methods. This paper is an important contribution to the discussion on how best to measure violence against women / gender based violence in all its forms via a survey. It highlighs a notable absence of standardised and validated tools for measuring GBVH in higher education institutions and lack of agreed definitions globally on the different forms of GBVH, including newer and emerging forms of violence such as technology facilitated gender based violence.
I enjoyed reading the paper and think it will add great value to the field.
I do however a few comments:
We are all striving to create measures of GBVH that are reliable and valid, have potential to be globally useful, are culturally sensitive, can be translated into other languages - to allow for comparisons across settings, yet flexible enough to enable use in other settings - which the ARC3 as the authors argue is not. However, what is missing from the paper is acknowledgement of the importance of context - and measures, like interventions - must be adapted to a local context. Yet, there is not mention in the paper of any attempt to adapt the ARC3 to a UK HEI context - if there was a pilot done or any adaptation done it would be important to mention it and if there wasn't this too should be mentioned, and why. And if no, the paper should make a strong recommendation on the importance of adaptation of measures.
One more point, I wonder why the team chose the ARC3 to use - did you do any review of tools and if yes, what criteria did you use to select the ARC3. There is no mention of the many existing measures (not specifically for HEI) that could have been adapted for this context - for example...WHO/DHS VAW IPV module;Measuring Gender Attitude: Using Gender-Equitable Men Scale (GEMS); The Social Norms and Beliefs about Gender Based Violence (GBV) Scale; International Men and Gender Equality Survey (IMAGES); Demographic and Health Surveys (searchable collection of DHS surveys); CARE’s rapid gender analysis; Acceptance of Modern Myths About Sexual Aggression (AMMSA); Inclusion of Other in Self Scale (for relationships closeness); The Hopkins Symptom Checklist (HSCL) (anxiety and depression scale); Specific questions with broader range of violence and controlling behavior (lots of comprehensive indicators, CDC US-based); Gender and Power Metrics (searchable collection of questionnaires used).
Many of the above will not be relevant, but some may be. There is little mention of work from LMICs that have measured campus sexual assault using the Sexual Experiences Survey–Short Form Version (SES-SFV). Although maybe not relevant to this paper, literature from beyond UK and USA should be sourced in the findings paper (alongwith recognition that other tools and measures exist outside UK and USA).
So no major comments on the paper. So no major comments on the paper. The only change I would like to see if a comment on the pilot/adaptation - and why ARC3.
Congratulations on a helpful contribution to the field.
Reviewer 3 Report
I commend the authors for their diligent research on domestic violence, stalking, and sexual violence. I appreciate their efforts in addressing these critical issues. However, I would like to offer some suggestions for improvement based on the following considerations:
1. Clarify Relationships Between Types of Violence:
While the authors note that domestic abuse and stalking in higher education (HE) have received less attention in research compared to sexual harassment and sexual violence, their study found that 26% of respondents experienced 'dating violence.' To enhance clarity, the paper should explicitly define the connections and interactions between "sexual violence" and domestic abuse and stalking in higher education (HE). This clarification is essential, especially considering that the context can vary significantly between countries, such as the USA and the UK, regarding stalking and 'dating violence.'
2. Separate Literature Review and Method:
I recommend separating the "literature review" and "Method and materials" sections into distinct entities to improve readability and understanding. Combining these two sections creates confusion within the article and disrupt its narrative flow.
3. Discuss Gender Differences:
The study included 446 men and 238 women as participants. However, it does not explore gender differences between men and women. It ignored it, as if it does not matter. What are the gender implications? A discussion of gender disparities or similarities would be valuable and engaging for many readers of the article when it is published. Hence, I encourage the authors to integrate insights from relevant published studies that discuss the gender dimension of data using student samples from the UK, such as the study referenced below:
Lazarus, S., Button, M., & Kapend, R. (2022). Exploring the value of feminist theory in understanding digital crimes: Gender and cybercrime types. The Howard Journal of Crime and Justice, 61(3), 381-398. https://doi.org/10.1111/hojo.12485
Examining how and why women may respond similarly or differently to men can be particularly insightful, especially when considering feminist perspectives.
4. Clarify Central Contribution:
The authors should provide a clearer explanation of the central contribution of the paper. Readers should understand what this paper contributes to the understanding of online self-presentation beyond the specifics of the case. Highlighting broader implications would add value to the study.
5. Strengthen Central Argument:
The central argument should be strengthened and streamlined. In articles of approximately 8,000 words, it is essential to have one central argument around which sub-themes and the entire paper revolve. More clarity is needed in this area.
6. Discuss Limitations:
It is important to discuss the limitations of the study clearly. This will help readers understand the scope and potential constraints of the research.
Minor Issue:
The "Discussion" section should begin with the number 4 rather than 3.1.3, which is a sub-section of the results.
The above suggestions will significantly help the authors enhance the research clarity, comprehensibility, and impact of the study.
Round 2
Reviewer 1 Report
I think you have improved deeply your work. Theoretical introduction and explanations about tools are excellent. So I can recomended now to be published.
Author Response
Thank you for your kind comments.
Reviewer 3 Report
The manuscript has been significantly improved and is now suitable for publication, in my opinion.
But see the minor issue to correct before publication:
Delete "and methods" from the text below since you have separated the "literature review" and "Method and materials" sections into distinct entities to improve readability and understanding, as I previously advised.
2. Literature review and methods
Careful proofreading is always needed to improve clarity.
Author Response
Thank you for your comment. This change has been made.